# Democrats are better than Republicans at discerning true and false news but do not have better metacognitive awareness

Mitch Dobbs[1], Joseph DeGutis[2,3], Jorge Morales [4,5], Kenneth Joseph[6] & Briony Swire-Thompson [1,7 ✉]

Insight into one's own cognitive abilities is one important aspect of metacognition. Whether this insight varies between groups when discerning true and false information has yet to be examined. We investigated whether demographics like political partisanship and age were associated with discernment ability, metacognitive efficiency, and response bias for true and false news. Participants rated the veracity of true and false news headlines and provided confidence ratings for each judgment. We found that Democrats and older adults were better at discerning true and false news than Republicans and younger adults. However, all demographic groups maintained good insight into their discernment ability. Although Republicans were less accurate than Democrats, they slightly outperformed Democrats in metacognitive efficiency when a politically equated item set was used. These results suggest that even when individuals mistake misinformation to be true, they are aware that they might be wrong.

[1] Network Science Institute and Department of Psychology, Northeastern University, Boston, MA, USA. [2] Boston Attention and Learning Laboratory, VA Boston Healthcare System, Boston, MA, USA. [3] Department of Psychiatry, Harvard Medical School, Boston, MA, USA. [4] Department of Psychology, Northeastern University, Boston, MA, USA. [5] Department of Philosophy, Northeastern University, Boston, MA, USA. [6] Department of Computer Science and Engineering, University at Buffalo, Buffalo, NY, USA. [7] Department of Political Science, Northeastern University, Boston, MA, USA. ✉email: b.swire-thompson@northeastern.edu

n today's information ecosystem, discerning between true and false information is an essential skill. Furthermore, being aware of one's ability to discern true from false statements is equally critical. Insight into one's own cognitive abilities is one important aspect of metacognition[1], and it is possible that people who have poor metacognitive insight are more likely to hold false beliefs[2]. Metacognition has only recently received attention in misinformation studies, and whether this construct varies across groups when discerning true and false information has yet to be examined. The current study examines how political partisanship, age, education, and gender influence (i) discernment ability, (ii) metacognitive efficiency (i.e., metacognitive ability given a specific level of task performance), and (iii) response bias (i.e., the tendency to answer true or false regardless of an item's veracity) for true and false news. Finally, we examine whether participants worse at detecting false news headlines show less insight into their discernment ability (i.e., a Dunning-Kruger effect[3]).

## Metacognition and discerning misinformation

People who are poor at discerning true and false information may be particularly prone to making poor decisions regarding health (e.g., foregoing cancer treatment or vaccination[4]), distrusting science[5], or abstaining from political participation[6]. Intuitively, individuals who mistakenly believe that they are good at detecting misinformation are more likely to hold false beliefs. As such, low metacognitive ability may explain why people believe and share misinformation, particularly in the absence of a correction or feedback. Unfortunately, individuals consistently assume that they are less susceptible to believing misinformation than other people[2,7,8]. This aligns with several findings that broadly describe people's tendency to evaluate themselves favorably relative to others (e.g., the third-person effect and the better-than-average effect[9–11]). Yet the question remains: Who has good awareness of their ability to separate fact from fiction?

Numerous studies have examined awareness of one's abilities across a variety of domains, finding that people generally show moderate levels of insight (e.g., subjective/objective $r \sim 0.30$[12]). However, studies explicitly investigating people's insight into their ability to separate true and false information are rare. One exception is Lyons et al.[2], where participants rated the accuracy of true and false news headlines, and then were asked how they compared to other Americans in their ability to recognize made-up news. The authors found that three in four participants overestimated their ability to detect made-up news, and that the lowest performers overestimated their ability the most. In turn, overestimation was associated with more frequent visits to low-quality websites and a greater willingness to share false content online. The authors also reported extremely low correlations between participants' perceived and actual ability to detect made-up news ($r = 0.08$ and $r = 0.10$), which fall below typical correlations between actual and perceived abilities. Similarly, Salovich and Rapp[7] asked participants to rate the veracity of statements from a story and found that participants overestimated their ability to detect inaccurate statements. This effect was also the most pronounced amongst the lowest performers on the task.

Although Lyons et al.[2] and Salovich and Rapp[7] suggest that participants have low insight into their ability to detect false information—particularly those worst at actually detecting it—there is also evidence that the opposite could be true. Fischer et al.[13] found that participants effectively discerned between true and false COVID-19 information, and also maintained good insight into this ability. On average, their metacognitive measure of interest approached its optimal value of 1 ($M_{m\text{-}ratio} = 0.86$[14,15]). This could be because this study assessed participants after each item, whereas Lyons et al.[2] and Salovich

and Rapp[7] asked participants to respond at the end of the experiment and to compare their performance against the general population. Finally, Arin et al.[16] also concluded that people had a good assessment of their ability to detect false news. However, they only compared whether participants would hypothetically share false news with whether they reported sharing false news in the past. In other words, their study focused on sharing rather than explicit measures of discernment ability.

## Demographic differences in discernment and metacognitive ability

Demographics such as partisanship, age, and gender have been previously related to discernment ability[17] and metacognition[18,19]. However, previous studies at the intersection of metacognition and misinformation have either not examined or not reported if results differed demographically. Demographic differences in discernment and metacognitive ability may enhance our understanding of why some groups (like political conservatives and older adults) engage more with low-quality news online[20,21]. Regarding partisanship, Republicans have generally been found to be worse than Democrats at discerning true and false news[13,22,23]. Garrett and Bond[17] suggested that it could be strong Republicans—but not strong Democrats—who perform particularly poorly on discernment measures, though some posit that both political extremes could be poor at the task[24]. While some research finds that both parties are better at discerning headlines congruent with their political ideology[25], it appears that Democrats are generally better at this task regardless of the content[22,23]. In terms of metacognitive ability, extreme partisans on both the political left and political right appear to overestimate the precision of their answers to political knowledge questions more than weaker partisans[26]. Furthermore, Rollwage, Dolan, and Fleming[19] found that, compared to those with less radical beliefs, participants holding more radical political beliefs exhibited lower metacognitive ability on a perceptual judgment task. As a result, it may be that stronger political partisans score lower on measures of metacognitive ability relative to weaker partisans.

For age, older adults are often better at discerning true from false news than younger adults[5,13,22], although some studies find no age differences in discernment[27,28]. By contrast, evidence suggests that metacognitive abilities generally decrease as people get older[18,29]. Older adults' confidence ratings also appear less predictive of their actual performance on both eyewitness identification and error awareness tasks[30,31]. Thus, older adults may score lower on measures of metacognitive ability compared to younger adults. Regarding education and gender, there are even fewer studies examining discernment and metacognitive differences. Intuitively, one might expect higher levels of education to predict higher metacognitive ability, as is the case with detecting misinformation[13,32,33]. However, there is also evidence that higher levels of education predict overconfidence and increased miscalibration between one's perceived and actual performance[34]. In terms of gender, limited evidence suggests that men might be more accurate than women at detecting fake news[13], while there is little evidence that men and women differ in metacognitive ability[35].

## Measuring discernment and metacognition

Previous findings regarding participants' insight into their ability to detect misinformation are limited by several methodological factors. For instance, Lyons et al.[2] and Salovich and Rapp[7] relied on single- or two-item measures of ability and asked participants to assess their performance relative to other Americans. Together, this may result in unreliable measurement[36,37] or

underestimating participants' insight into their own ability[38–40]. Studies by Fischer and colleagues[14,41] used a more robust method of measuring participants' self-insight over a greater number of trials ($n = 10$ and 33, respectively), providing slightly more reliable estimates of metacognitive ability. A notable strength of these studies is their use of the *meta-d'* framework to measure metacognitive ability[15,16]. This framework has several advantages over and above approaches used in other misinformation studies, including minimizing performance and response bias artifacts and quantifying discernment ability and response bias separately[42]. However, these studies had low trial counts, and utilizing a larger number of trials would further improve the reliability and precision of their measures[43].

### The current study

The current study examined individuals' insight into their ability to separate true and false news headlines, and how this varies with political partisanship, age, education, and gender. To do so, we utilized a signal detection theory-based model (SDT[16,44]) that independently measures discernment ability, metacognitive ability, and response bias. We preregistered three hypotheses: (i) stronger political partisans will score lower on metacognitive ability than weaker partisans[19,26]; (ii) younger adults will score higher than older adults on metacognitive ability[18,29]; and (iii) higher levels of education will predict higher discernment ability[32,33]. We also include several exploratory research questions regarding (i) how participants' performance on discernment, metacognitive ability, and response bias vary according to the political favorability of our headlines and (ii) whether participants who score lower on discernment also score lower on metacognitive ability[2,7].

### Methods

**Participants**. We recruited 533 participants using Prolific. We recruited participants so that there were equal numbers of men, women, Democrats, and Republicans in each age bin (18–32, 33–47, 48–62, 63+). We chose these age bins to replicate previous work showing that people aged 18-30 differed in comparison to people aged 65+ in fake news engagement online[20,21]. Our a priori exclusion criteria were participants who reported a lack of effort ($N = 5$) and did not answer all items ($N = 15$). We used the outlier labeling rule to exclude participants with extreme metacognitive efficiency and discernment values ($N = 9$)[45]. Finally, we removed participants with negative *m-ratio* values ($N = 4$)[43]. In our final sample ($N = 500$), there were 247 men, 252 women, and 1 individual who chose not to disclose their gender. Participants' age ranged from 18 to 84 ($M = 47.2$, $SD = 16.5$). Note that we also excluded participants with negative or extreme *m-ratio* values when re-estimating metacognitive efficiency for our politically equated stimulus set (where we removed $n = 9$ additional participants; $N = 491$), and when calculating metacognitive efficiency for the political favorability analyses (where we removed $n = 40$ participants; $N = 460$).

**Sample size justification**. Studies analyzing metacognitive differences involving psychophysical tasks typically use small sample sizes and a larger number of trials. For instance, Rahnev et al.[46] compiled 145 datasets investigating confidence measurements and found that the median sample size was 37 participants over 309 trials. Given that exact effect sizes for metacognitive differences in news discernment tasks remain unknown, we based our sample size on studies that also investigate metacognition and/or discernment ability. For instance, Fischer, Amelung, and Said[41] had 509 participants and 8 trials, Scott et al.[47] had 450 participants and 60–64 trials, and Sultan et al.[48] used 760 participants

and 37 trials (for news headlines). Thus, we decided that 500 people and 140 trials would be sufficient, giving us 125 participants in each of our age bins. This sample size is greater than other known studies comparing older and younger adults (for instance, $N = 60$[18] and $N = 72$[49]).

A sensitivity analysis in G*Power[50] indicated that our final sample size of 500 people had 95% power to detect the outcomes of a between-subjects ANOVA for our partisanship × partisanship strength analyses on discernment and metacognitive ability of at least $f = 0.19$, and a one-way ordinal ANOVA with age group as a factor at $f = 0.19$. This aligns with recommendations by Brysbaert[51], recommending that $f = 0.20$ is a good estimate for the smallest effect size of interest in psychological research. However, we acknowledge that we may be underpowered to detect smaller effect sizes. We therefore repeat all analyses with Bayesian methods, reporting all null findings with the relative evidence favoring the null ($BF_{01}$) quantified in the main text. For reference, a BF between 1 and 3 provides anecdotal evidence, 3–10 moderate evidence, 10–30 strong evidence, 30-100 very strong evidence, and a BF greater than 100 constitutes extreme evidence[52].

**Procedure**. Ethical approval for this study was granted by the Northeastern University IRB (#19-04-09). Participants first provided informed consent and then rated all fact and misinformation items in a randomized order. Participants reported (i) whether or not they believed each item to be true (Yes/No), and (ii) how confident they were in their choice using a four-point scale (1 = Not Confident, 2 = Barely Confident, 3 = Somewhat Confident, 4 = Very Confident). Finally, participants answered demographic questions (i.e., self-reported age, political affiliation, education level, and gender). Participants were paid $13.70 per hour.

**Stimuli**. To enhance the temporal validity of our stimuli, we ensured that all news headlines pertained to topics reported within one year of data collection (August 17th, 2022). We compiled a list of items from previous studies[53,54], and adapted false items from third-party fact-checking websites (e.g., Snopes and PolitiFact). True stimuli were adapted headlines from a variety of mainstream sources (e.g., CNN, NPR, Fox News). All headlines were shortened and paraphrased for clarity, and false claims that were not originally headlines were phrased to read like a headline (see Supplementary Table 1). This produced 100 true and 100 false items. We ran a pilot study to choose our final items (see Supplementary Methods). Our final stimuli set contained 70 true ($M = 3.02$, $SD = 1.24$) and 70 false items ($M = 3.36$, $SD = 1.25$; see Supplementary Figure 1).

This stimulus set was not perfectly balanced for political favorability, with more false items favorable to Republicans ($n = 54$) than Democrats ($n = 16$), and more true items favorable to Democrats ($n = 39$) than Republicans ($n = 31$). Although this resembles the nature of the current media ecosystem[17,55], we repeated all analyses using a politically equated version of this stimulus set. This was a necessary check because we did not want Democrats who endorsed all the pro-Democrat items performing well, and Republicans who endorsed all pro-Republican items (which contained more misinformation) performing poorly. We exclusively used the politically equated stimulus set for analyses that directly examined biases (such as political favorability), as we were interested in how Democrats and Republicans performed on items that are equally consistent and counter to their worldviews.

To ensure that our performance comparison was as fair to both parties as possible, our equated stimuli set had 64 items, 16 true and 16 false items favorable Democrats ($M_{True} = 2.62$, $SD_{True} =$

**Table 1 Definitions for signal detection theory response classifications.**

| Classification | Definition |
| --- | --- |
| Type 1 Hit | True trials classified as true |
| Type 1 Miss | True trials classified as false |
| Type 1 Correct Rejection | False trials classified as false |
| Type 1 False Alarm | False trials classified as true |
| Type 2 Hit | Type 1 hits or correct rejections answered with high confidence |
| Type 2 Miss | Type 1 hits or correct rejections answered with low confidence |
| Type 2 Correct Rejection | Type 1 misses or false alarms answered with low confidence |
| Type 2 False Alarm | Type 1 misses or false alarms answered with high confidence |

0.13; $M_{\text{False}} = 2.59$, $SD_{\text{False}} = 0.33$), and 16 true and 16 false items favorable to Republicans ($M_{\text{True}} = 3.38$, $SD_{\text{True}} = 0.36$; $M_{\text{False}} = 3.39$, $SD_{\text{False}} = 0.48$). Finally, we included five extra items in order to investigate how much variance in discernment was predicted by a newly developed scale, the Misinformation Susceptibility Test (MIST; see Supplementary Results[54]).

**Analysis plan.** We conducted all frequentist analyses and modeling in R[56], and all Bayes Factor calculations in JASP[57]. To measure participants' discernment ability and response bias, we utilized SDT. SDT uses binary judgments (e.g., true/false) and confidence ratings to calculate type-1 and type-2 hits and false alarms (see Table 1). Type-1 judgments pertain to the accuracy of binary judgments classifying stimuli (e.g., the truth value of a statement), whereas type-2 judgments refer to how well confidence ratings discriminate between the subject's correct and incorrect type-1 classifications. Using type-1 rates, discernment ability ($d'$) can be estimated, where discernment ability (or discrimination sensitivity) refers to how effectively participants distinguish between two classes of stimuli (see Supplementary Methods).

Type-1 judgments can also be used to estimate participants' response bias (i.e., $c$ value), or tendency to provide the same binary judgment across all trials. In this study, $c$ values can be interpreted as the amount of "trueness" (i.e., signal) a participant must observe in order to rate a headline as true. Positive $c$ value indicates a strict threshold for rating a headline as true: Because this criterion is strict, participants with a positive $c$ value will rate more headlines as false. The opposite is true for participants with a negative $c$ value: Because their criterion for rating a headline as true is more relaxed, they will produce more true responses. Importantly, participants' criterion for rating a particular headline as true is calculated independently from their ability to discern between true and false headlines. The ability to treat these constructs as distinct is a major advantage of SDT. We also calculated a $c'$ (i.e., "c prime") value for each participant, which represents their response bias relative to their discernment ability (i.e., their $d'$ value; see Supplementary Results).

To measure metacognitive ability, we use Maniscalco and Lau's *meta-d'* approach[15,58]. The *meta-d'* model leverages SDT to estimate type-2 hit and false alarm rates from participants' confidence ratings. In this study, confidence was rated on a four-point scale. Ratings of three or four were considered high confidence, and confidence ratings of one or two were considered low confidence. This enables a type-2 parallel to $d'$ to be computed: *meta-d'*. Meta-d' can be interpreted as the $d'$ value that a participant would need to be considered metacognitively ideal (i.e., expressing high confidence in every hit, low confidence in every false alarm, etc.) based on their confidence ratings.

Unlike alternative measures of metacognitive ability, the *meta-d'* model separates how well a participant makes metacognitive judgments from their metacognitive bias[42]. Additionally, *meta-d'* is measured in $d'$ units, meaning the two terms can be directly compared. Simply dividing *meta-d'* by $d'$ minimizes type-1

performance artifacts, producing a measure of metacognitive efficiency called the *m-ratio*. M-ratio values can be interpreted as a participant's metacognitive ability given a specific level of task performance, or how metacognitively capable (efficient) one is given how difficult one finds the task. A metacognitively ideal observer would produce a *meta-d'* value equivalent to their $d'$ value, yielding an *m-ratio* of 1. We focus on *m-ratio* values as a dependent measure in this study as opposed to *meta-d'* values because the former accounts for how participants actually performed on the task.

To estimate *meta-d'*, we deviate from our preregistration and implement a non-hierarchical Bayesian method[16]. This approach is better able to quantify uncertainty in parameter estimates and more effectively handles cells with zero counts, unlike maximum-likelihood/sum of squared error approaches[44,59]. Bayesian methods also allow evidence to be collected in favor of the null hypothesis and combine prior information with new data[52]. We ran three Markov Chain Monte Carlo (MCMC) chains with 10,000 samples each via JAGS (see https://mcmc-jags.sourceforge.io/), with the first 2,000 discarded as warm-up. We used the priors specified by Fleming[16] for individual estimates of *meta-d'* ($d' \sim Normal(0, 0.5)$; $c \sim Normal(0, 2)$; *meta-d'* $\sim Normal(d', 0.5)$, and assessed model convergence by calculating Gelman-Rubin statistics[60]. These values indicated good convergence across chains (all $\hat{R} < 1.01$). We deviate from our preregistration and use the raw *meta-d'* values generated by the model and not their logarithm, since we report *m-ratio* values as our metacognitive measure of interest. For robustness, we also estimated *meta-d'* hierarchically for all partisanship analyses, as this methods better handles smaller numbers of trials[16].

**Pre-registration.** The following analyses were pre-registered (see https://osf.io/ay9fc/), and we label all exploratory analyses below. We preregistered the hypotheses that stronger political partisans would score lower on metacognitive ability than weaker partisans[19,26]; younger adults would score higher than older adults on metacognitive ability[18,29]; and higher levels of education would predict higher discernment ability[32,33]. Note that we pre-registered two additional hypotheses for discernment: that stronger political partisans would score lower than weaker partisans, and that older adults would score lower than younger adults. However, we subsequently realized that this did not align well with the current literature, and predictions of Democrats and older adults being better at discernment would have been better aligned.

**Reporting summary.** Further information on research design is available in the Nature Portfolio Reporting Summary linked to this article.

## Results

We report the following analyses using all 140 items presented to participants ($n = 70$ false items; $n = 70$ true items; see Methods).

This stimulus set contained slightly (but significantly) more false news favorable to Republicans (see Supplementary Figure 1), which resembles the current misinformation ecosystem[17,55]. As a robustness check, we repeated all analyses using a stimulus set equated for political favorability (see Methods; Supplementary Table 1) and note the few effects that differed from the full dataset. For analyses that directly examine biases (response bias and political favorability), we report findings from the politically equated item set alone. We include Bayes Factors for all null results throughout the manuscript. Examining the consistency of the results across these multiple analytic approaches reduces the likelihood of making type-I errors (akin to sensitivity analyses[61,62]). All statistical tests were two-tailed. Distributions were tested for normality using Kolmogorov-Smirnov and Shapiro–Wilk tests. In cases when the data were not normally distributed, all findings replicated when using non-parametric tests.

**Discernment ability**. We first sought to examine participants' ability to discriminate between true and false statements and how this varied with partisanship, age, education, and gender. Overall group accuracy was 78.9% for facts and 81.3% for false statements, producing an average $d'$ of 1.82 ($SD = 0.62$; see Supplementary Figure 2 for correct responses split by confidence rating). Next, we conducted a 2 × 2 factorial ANOVA with between-subjects factors partisanship (Republican vs. Democrat) and partisanship strength (strong vs. weak) on $d'$ values. We found a significant main effect of partisanship ($F(1, 494) = 172.87$; $p < 0.001$; $MSE = 0.26$; $\eta p^2 = 0.26$), showing that Democrats were more accurate than Republicans. This was qualified by a partisanship × partisanship strength interaction on $d'$ values ($F(1, 494) = 31.99$; $p < 0.001$; $MSE = 0.26$; $\eta p^2 = 0.06$), indicating that partisanship's influence on $d'$ varied with partisanship strength. As can be seen from Fig. 1, weak Republicans were more discerning than strong Republicans $t(235) = 2.75$, $p = 0.007$, 95% CI = [0.049, 0.297], Cohen's $d = 0.35$), but strong Democrats were more discerning than weak Democrats ($t(129) = 4.71$, $p < 0.001$, 95% CI = [0.210, 0.516], Cohen's $d = 0.69$). This same pattern was found using politically equated stimuli.

Turning to age, we conducted a one-way ordinal ANOVA with age group as a factor (18–32, 33–47, 48–62, 63+) on $d'$ values. This revealed a significant main effect ($F(3, 496) = 2.72$; $p = 0.044$; $MSE = 0.38$; $\eta p^2 = 0.02$), illustrating that discernment ability differed with age. However, we note that the effect of age did not cross the threshold of significance when repeated using politically equated items ($F(3, 487) = 2.60$; $p = 0.052$; $MSE = 0.32$; $\eta p^2 = 0.02$; $BF_{01} = 3.77$). Planned comparisons on the full stimuli set revealed that older adults (63+) had significantly higher $d'$ scores than younger adults (18–32; $t(247) = 2.35$, $p = 0.019$, 95% CI = [0.029, 0.330], Cohen's $d = 0.29$). There were no statistically significant differences between $d'$ scores for the 33–47 and 48–62 age groups ($t(244) = 0.44$, $p = 0.663$, 95% CI = [−0.123, 0.193], Cohen's $d = 0.06$; $BF_{01} = 6.56$). For robustness, we also correlated $d'$ with age, and found that they were positively associated, albeit modestly ($\rho = 0.11$, $p = 0.016$).

Finally, we investigated whether discernment ability was related to education and gender. We correlated $d'$ with level of education, and found that more educated participants were more successful at separating true from false headlines ($\rho = 0.25$, $p < 0.001$). To examine gender differences in $d'$ scores, we conducted an independent samples $t$-test and found that men in our sample had higher $d'$ values than women ($t(496) = 4.20$, $p < 0.001$, 95% CI = [0.123, 0.338], Cohen's $d = 0.38$). When considering these demographic factors together in a multiple regression predicting discernment ability ($d'$), we found that each of these factors predicted unique variance in truth discernment

(partisanship: $\beta = 0.52$, $SE = 0.05$, $p < 0.001$; education $\beta = 0.19$, $SE = 0.02$, $p < 0.001$; gender: $\beta = 0.18$, $SE = 0.05$, $p < 0.001$; age: $\beta = 0.07$, $SE = 0.001$, $p = 0.039$), with a total model predicting 36% of the variance in discernment ability. Note that when repeating this analysis with politically equated items, the significant predictors were identical except that age did not reach significance ($\beta = 0.08$, $SE = 0.001$, $p = 0.057$; $BF_{01} = 2.61$).

**Metacognitive efficiency**. We next sought to examine our metacognitive measure of interest, metacognitive efficiency (i.e., $m$-ratio values), and how this varied across partisanship, age, education, and gender (see Supplementary Results for $meta$-$d'$ results). $M$-ratio values can be interpreted as how aware one is of their ability to discern true from false news when controlling for their actual performance. The average $m$-ratio value for our sample was high ($M = 0.86$, $SD = 0.29$), indicating that participants were generally metacognitively efficient. As can be seen in Fig. 2, there were no significant differences in metacognitive efficiency across partisanship (neither its main effect, $BF_{01} = 5.84$, or interaction with partisanship strength, $BF_{01} = 4.57$), age ($BF_{01} = 8.07$), gender ($BF_{01} = 6.05$), or education level ($BF_{01} = 17.84$; all $ps > 0.218$).

When repeating these analyses with the stimulus set equated for political favorability, we found a main effect of partisanship ($F(1, 485) = 4.26$; $p = 0.039$; $MSE = 0.12$; $\eta p^2 = 0.01$), and a main effect of partisanship strength ($F(1, 485) = 5.31$; $p = 0.022$; $MSE = 0.12$; $\eta p^2 = 0.01$), though no partisanship × partisanship strength interaction ($F(1, 485) = 0.09$; $p = 0.762$; $MSE = 0.12$; $\eta p^2 = 0.00$; $BF_{01} = 6.28$). As seen in Supplementary Figure 3, Republicans had slightly higher metacognitive efficiency than Democrats using equated items, and stronger partisans appear more metacognitively efficient than weaker partisans.

**Response bias**. Next, we turned to examining variation in response bias (i.e., $c$ values) across demographics (see Supplementary Results for $c'$ analyses). Note that we used the politically equated stimulus set for this analysis (see Methods). When conducting a 2 × 2 factorial ANOVA with between-subjects factors partisanship (Republican vs. Democrat) and partisanship strength (strong vs. weak) on $c$ values, we found a main effect of partisanship ($F(1, 485) = 5.17$; $p = 0.023$; $MSE = 0.12$; $\eta p^2 = 0.01$). As can be seen from Supplementary Figure 4, Democrats had a slightly greater "false bias" than Republicans. However, there was no evidence of statistically significant differences in age when conducting a one-way ordinal ANOVA ($F(3, 487) = 0.61$, $p = 0.610$, $MSE = 0.12$, $\eta p^2 = 0.004$, $BF_{01} = 55.52$) or correlating age as a continuous measure with $c$ values ($\rho = -0.03$, $p = 0.459$, $BF_{01} = 13.93$). Finally, we found no statistically significant evidence of differences in response bias across education ($\rho = -0.05$, $p = 0.266$, $BF_{01} = 6.12$) or gender ($t(488) = 0.36$, $p = 0.720$, 95% CI = [−0.051, 0.074], Cohen's $d = 0.03$, $BF_{01} = 9.36$).

**Political favorability of the news headlines**. We next performed exploratory analyses to investigate whether the political congruence of our items influenced participants' performance. Again, we conducted these analyses using the politically equated stimulus set, as we were interested in how Democrats and Republicans performed on items that were equally consistent and counter to their worldviews.

To analyze differences in discernment ability across pro-Democrat and pro-Republican items, we conducted a 2 × 2 between-within ANOVA with between-subjects factors partisanship (Democrat vs. Republican) and within-subjects factor item type (pro-Republican vs. pro-Democrat items). We found a main

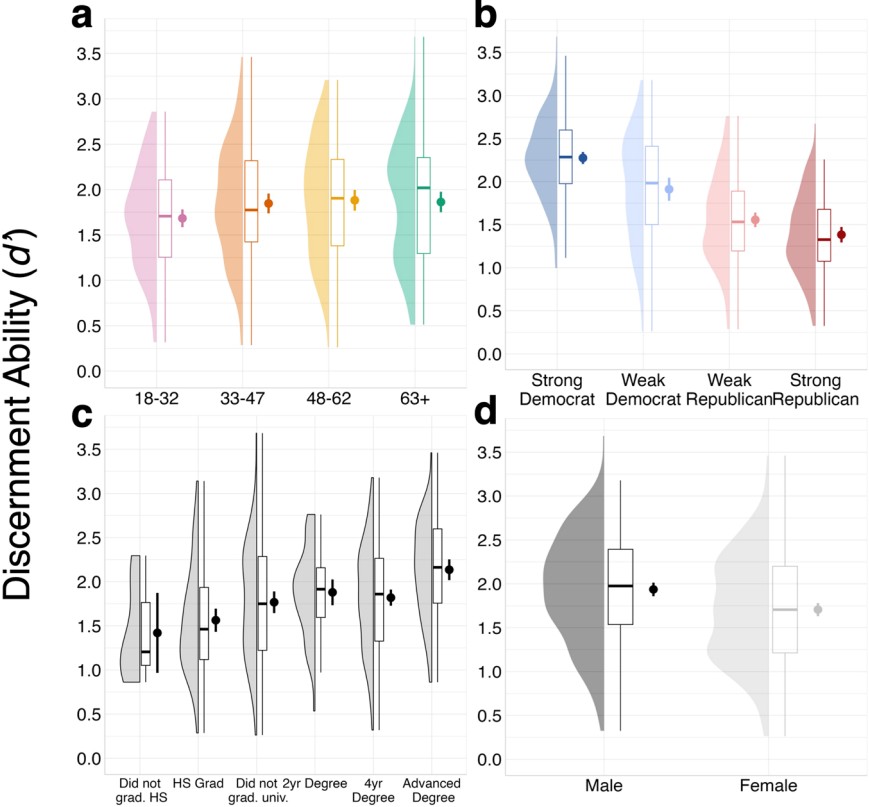

**Fig. 1 Demographic distributions for discernment ability.** Distribution, boxplot, and point estimate of *d'* scores for **a** ages 18–32 (pink; $n = 125$), 33–47 (orange; $n = 125$), 48–62 (yellow; $n = 122$), and 63+ (green; $n = 128$). **b** strong Democrats (dark blue; $n = 166$), weak Democrats (light blue; $n = 84$), weak Republicans (light red; $n = 144$), and strong Republicans (dark red; $n = 104$). **c** education levels (left to right: $n = 6, 81, 109, 47, 176, 81$). **d** men (dark gray; $n = 247$) and women (light gray; $n = 252$). Error bars reflect 95% confidence intervals, boxplot midlines reflect the median, and point estimates reflect the mean in all panels.

effect of partisanship ($F(1, 458) = 68.99$; $p < 0.001$; $MSE = 0.49$; $\eta p^2 = 0.13$), indicating that Democrats were more accurate than Republicans. We also found a partisanship × item type interaction on *d'* values ($F(1, 458) = 5.12$; $p = 0.024$; $MSE = 0.25$; $\eta p^2 = 0.01$), driven by partisan performance varying with the political favorability of the headline. As seen in Fig. 3, Democrats outperformed Republicans to a greater extent on pro-Republican items than pro-Democrat items (see Supplementary Figure 5 for the data split by partisanship and partisanship strength).

To examine whether metacognitive efficiency differed according to the political favorability of our items, we repeated the 2 × 2 between-within ANOVA on *m-ratio* values with between-subjects factors partisanship and within-subjects factor item type. We found a main effect of partisanship ($F(1, 458) = 5.36$; $p = 0.021$; $MSE = 0.17$; $\eta p^2 = 0.01$), showing that Republicans were *more* metacognitively efficient than Democrats, although we did not find an interaction between partisanship and item type ($F(1, 458) = 3.28$; $p = 0.071$; $MSE = 0.13$; $\eta p^2 = 0.01$, $BF_{01} = 1.95$). For robustness, we also estimated *meta-d'* at the group level (i.e., hierarchically) as this method is more robust to smaller numbers of trials[16]. Using this method, we found no statistically significant evidence of differences in metacognitive efficiency between Republicans and Democrats on either item type (see Supplementary Figs. 6–7 and Supplementary Note 1).

Finally, we conducted a 2 × 2 between-within ANOVA with between-subjects factors partisanship and within-subjects factor item type on *c* values. We found a main effect of item type ($F(1, 458) = 359.04$; $p < 0.001$; $MSE = 0.06$; $\eta p^2 = 0.44$), reflecting that participants were more likely to exhibit a false bias for items favorable to Democrats relative to items favorable to Republicans

($t(459) = 17.7$, $p < 0.001$, 95% CI = [0.260, 0.324], Cohen's $d = 0.77$). Additionally, we found a partisanship × item type interaction on *c* values ($F(1, 458) = 52.68$; $p < 0.001$; $MSE = 0.06$; $\eta p^2 = 0.10$), showing that partisan response bias differed across pro-Democrat and pro-Republican items. As can be seen in Fig. 3c, for pro-Democrat items, both Democrats and Republicans showed a "false" bias (but they did not significantly differ from each other; $t(451) = 1.76$, $p = 0.079$, 95% CI = [−0.007, 0.126], Cohen's $d = 0.16$; $BF_{01} = 2.16$). However, for pro-Republican items, Democrats showed little-to-no bias, whereas Republicans showed a "true" bias ($t(424) = 4.57$, $p < 0.001$, 95% CI = [0.096, 0.240], Cohen's $d = 0.43$).

**Comparing the metacognitive performance of high and low performers.** Finally, we investigated whether the least accurate participants maintained the least insight into their discernment ability. We split participants into four quartiles based on their *d'* scores, and conducted a one-way ordinal ANOVA with factor quartile on *m-ratio* values. We found a significant main effect ($F(1, 471) = 24.53$; $p < 0.001$; $MSE = 0.06$; $\eta p^2 = 0.05$), showing that *m-ratio* varied with discernment ability. When comparing the bottom quartile to the top quartile, we found that the least accurate participants maintained *the most* insight into their ability ($t(203) = 4.60$, $p < 0.001$, 95% CI = [0.087, 0.218], Cohen's $d = 0.60$). For robustness, we split the data into even and odd trials, recalculated *d'* and *m-ratio* values, and repeated the one-way ordinal ANOVA to compare *m-ratio* values across quartiles. There was no main effect of quartile on even *d'* trials predicting odd *m-ratio* trials ($F(1, 471) = 0.55$, $p = 0.461$, $MSE = 0.12$, $\eta p^2 = 0.001$, $BF_{01} = 7.53$) or odd *d'* trials predicting

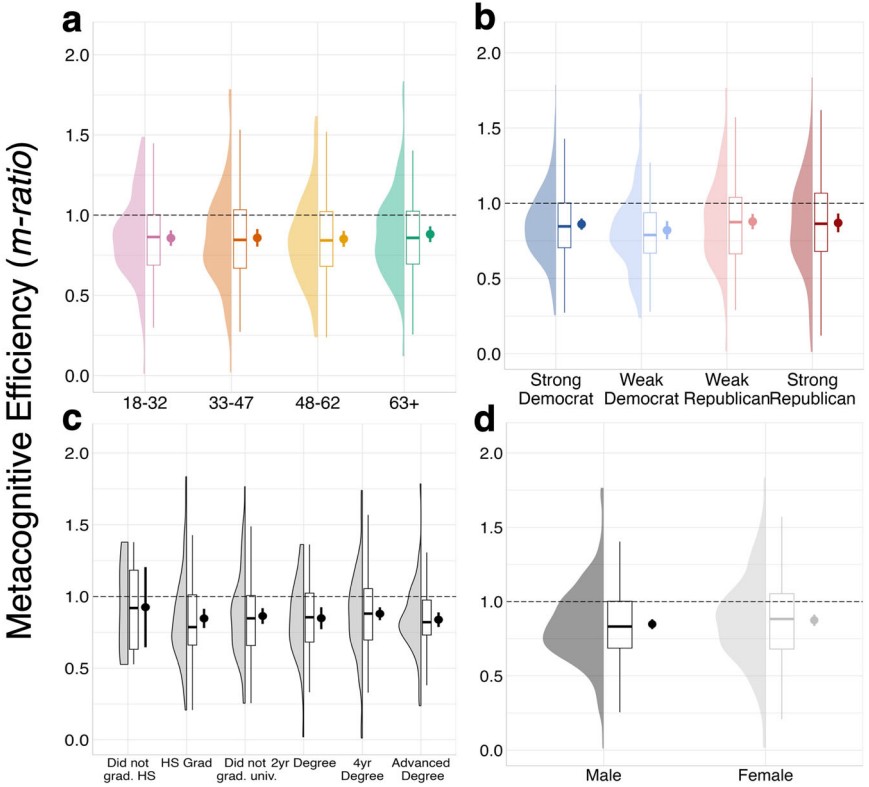

**Fig. 2 Demographic distributions for metacognitive efficiency.** Distribution, boxplot, and point estimate of *m-ratio* scores for **a** ages 18-32 (pink; $n = 125$), 33–47 (orange; $n = 125$), 48-62 (yellow; $n = 122$), and 63+ (green; $n = 128$), **b** strong Democrats (dark blue; $n = 166$), weak Democrats (light blue; $n = 84$), weak Republicans (light red; $n = 144$), and strong Republicans (dark red; $n = 104$), **c** education levels (left to right: $n = 6$, 81, 109, 47, 176, 81), and (**d**) men (dark gray; $n = 247$) and women (light gray; $n = 252$). Error bars reflect 95% confidence intervals, boxplot midlines reflect the median, and point estimates reflect the mean in all panels.

even *m-ratio* trials ($F(1, 471) = 1.46$; $p = 0.227$; $MSE = 0.11$; $\eta p^2 = 0.003$; $BF_{01} = 4.82$; see Supplementary Fig. 8 and Supplementary Note 2). Although this is inconsistent with the finding that the lowest-scoring performers had the highest metacognitive efficiency, it does not support the finding that the least discerning participants have the *poorest* metacognitive efficiency.

## Discussion

The current study investigated participants' judgments of true and false news headlines, examining differences in discernment ability, metacognitive efficiency, and response bias across demographics. When discerning between true and false news headlines, we found that Democrats performed better than Republicans, older adults performed better than younger adults, and men performed better than women. Furthermore, education was positively associated with discernment ability. Importantly, we found little-to-no differences in metacognitive efficiency in our sample. In other words, regardless of actual performance, people across all demographic groups maintained good insight into their discernment ability: The individuals who performed well knew they performed well, and individuals who performed poorly knew that they performed poorly. There were also few differences with respect to participants' bias to respond true or false. Although Democrats were slightly more likely than Republicans to rate headlines as false, participants' propensity to evaluate a headline as true or false did not differ with age, education, or gender.

On the whole, these findings illustrate that, individuals were good at discerning our true and false news headlines. Regarding partisanship, Democrats were more accurate at discerning news

veracity than Republicans[13,22,23]. We found that strong Republicans were worse at discernment than weak Republicans, but strong Democrats performed better than weak Democrats, aligning with previous research[17]. One potential reason for this is that strong partisans on both sides are more likely to engage with a greater quantity of news, but Democrats are more likely to engage with a greater *breadth* of news sources than Republicans[63], contributing to different prior beliefs about what is or is not plausible[22]. Republicans were also worse than Democrats at discerning news veracity regardless of whether or not the headlines were congruent with their political ideology[22,23]. Additionally, our results align with a growing body of evidence that older adults can detect fake news better than younger adults[5,13]. However, it is important to note that our effect sizes are quite small and might be specific to the current paradigm, and should be replicated. Furthermore, future studies should consider why such age differences potentially exist (e.g., differences in emotional processing[64,65]), particularly considering that older adults share more misinformation online than younger adults[66].

We also found that participants had good insight into their ability to detect false news, with minimal partisan differences in metacognitive efficiency. Although Republicans were less accurate than Democrats, they had good insight into their abilities, even slightly outperforming Democrats on metacognitive efficiency when using equated items. This speaks against the idea that Republicans confidently endorse misinformation: When they labeled misinformation as true, they were aware they might be wrong. Essentially, Republicans in our sample do not demonstrate "overconfidence" when assessing the veracity of news headlines. We also found that older adults performed well on measures of

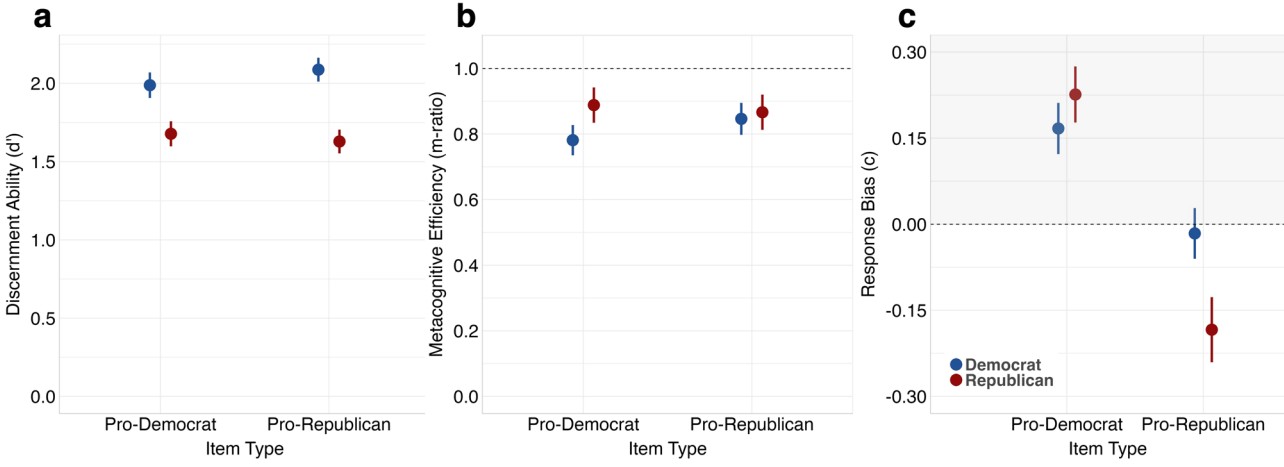

**Fig. 3 $d'$, $m$-ratio, and $c$ values split by political party and item favorability. a** Mean point estimates of discernment ability. **b** Mean point estimates of metacognitive efficiency. The dotted line at $y = 1$ represents optimal metacognitive efficiency. **c** Mean point estimates of response bias. Points in the gray area represent a tendency to answer false, and points in the white area represent a tendency to answer true. Estimates for Democrats shown in blue ($n = 239$) and Republicans in red ($n = 221$). Error bars reflect 95% confidence intervals in all panels.

metacognitive efficiency, casting doubt on the explanation that they share disproportionately more misinformation online because of a lack of insight into their abilities.

Regarding response bias, both Democrats and Republicans maintained a slight tendency to rate a given headline as false. However, Republicans exhibited a "true bias" (a tendency to evaluate items as true) for pro-Republican items, but a "false bias" (a tendency to evaluate items as false) for pro-Democrat items. This result suggests that Republicans' lower discernment ability could be at least partially driven by a tendency to answer politically congruent items as true (even on equated items[25]; though see Pennycook and Rand[22]). By contrast, Democrats demonstrated a false bias for pro-Democrat items and little-to-no bias for Republican items (counter to Batailler et al.[25]). This finding should be replicated prior to making firm conclusions, but it certainly appears that Democrats are not behaving in a way that aligns with traditional motivated reasoning accounts[67]. Finally, there were no significant differences in response bias between age groups, suggesting that gullibility (i.e., believing most news to be true) is not a fundamental mechanism underlying why older adults share disproportionately more false news online.

Our results also challenge the finding that those least capable of discerning true and false information (i.e., the least accurate individuals), are also the least aware of their abilities. We found that the worst performers in our sample *did not* have the worst metacognitive efficiency. One reason the current findings differ from Lyons et al.[2] and Salovich and Rapp[7] is that these studies asked participants to rate their abilities (i) after completing the task, and (ii) in comparison to the general population (i.e., other Americans), whereas we asked participants to rate their confidence after every trial. In other words, poorer performers may have insight into how they performed on the trial that just occurred, but lack awareness of how they did on the task as a whole and in comparison to the general population. Another possible reason that we did not observe typical Dunning-Kruger effects is because our discernment task was relatively easy, and miscalibration has been shown to increase with the difficulty of the task[68]. To further explore these possibilities, future studies should examine both trial-by-trial (i.e., local confidence in single decisions) and broader evaluations of perceived performance (i.e., global estimates[69]).

## Limitations
Although the current results are compelling, they are limited by several factors. First, our larger stimuli set was not balanced for

political favorability between true and false items, containing more false news favorable to Republicans. Although this is more ecologically valid with respect to the current information ecosystem[54,70], and we replicated our analyses with an equated stimulus set, this may have influenced overall response bias. Second, there may be limits to the generalizability of our task, given that participants saw an equal number of true and false news items, whereas in the real world, an overwhelming majority of the content people see is true[71]. It would be interesting to repeat this study using proportions of items more congruent with current news content. Third, older adults in online recruitment platforms may differ from the general population (though comparisons between online and offline studies suggest this difference may be minimal[72]). Finally, we did not recruit political independents or non-partisans, despite the fact that a large proportion of U.S. adults identify as such[73].

The intersection of metacognition and misinformation is a growing area of research, and the current study shows that applying SDT approaches can improve measurement and provide new insights[74–76]. This study finds that individuals—across all demographics—have good awareness of their ability to discern between true and false news, and speaks against metacognitive ability as a general mechanism driving demographic differences in the endorsement and spread of online misinformation. Instead, our results suggest that both discernment ability and response bias may drive engagement with misinformation, particularly on the political right. It remains to be seen whether metacognitive awareness could be used as an intervention approach to boost people's sharing decisions online or encourage individuals to examine dubious information further. Nonetheless, it is extremely hopeful that all demographic groups maintained good insight into their discernment ability, and that even when individuals mistake misinformation to be true, they are aware that they might be wrong.

## Data availability
Data necessary to replicate these analyses is available at https://osf.io/ay9fc/.

## Code availability
All code necessary to replicate the results reported in this manuscript is available via https://osf.io/ay9fc/.

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

## Acknowledgements

We acknowledge financial support from the Volkswagen Foundation: Grant "Reclaiming Individual Autonomy and Democratic Discourse Online" and NSF 2137871 "A Disinformation Range to Improve User Awareness and Resilience to Online Disinformation", both awarded to BST. The funders had no role in study design, data collection and analysis, decision to publish or preparation of the manuscript. We also thank Kristen Kilgallen for assistance in replicating analyses.

## Author contributions

M.D., J.D., J.M., and B.S.T. designed the study. B.S.T. and M.D. recruited participants. M.D. conducted the data analysis and M.D. and B.S.T. drafted the manuscript. B.ST., J.D., J.M., and K.J. provided critical revisions. All authors approved the final version of the manuscript for submission.

## Competing interests

The authors declare no competing interests.
