## [Peer Review File · Communications Psychology]

31st Jul 23

Dear Dr Swire-Thompson,

We greatly apologise for the delay in processing your manuscript. Thank you for your patience!

Your manuscript titled "Democrats are better than Republicans at discerning true and false news but do not have better metacognitive awareness" has now been seen by 3 reviewers, and I include their comments at the end of this message. They find your work of interest, but raised some important points. We are interested in the possibility of publishing your study in *Communications Psychology*, but would like to consider your responses to these concerns and assess a revised manuscript before we make a final decision on publication.

We therefore invite you to revise and resubmit your manuscript, along with a point-by-point response to the reviewers. Please highlight all changes in the manuscript text file.

The reviewers request more detail, further clarifications and justification for your predictions and design, as well as the inclusion of some additional literature. Reviewer #2 asks for further support of the null results. The journal has strong policies for the reporting and interpretation of null results, as detailed below. Please report Bayes Factors for all null results (currently included in the SI) in the manuscript. As you revise for clarity, please note that we ask for the following order of sections: Introduction - Methods - Results - Discussion.

To comply with the journal's statistics guidelines, please remove any interpretation of null effects obtained through null hypothesis significance testing.

Null results obtained in NHST should be mentioned as "yielding no statistically significant evidence" for an effect/a difference.

For example, you cannot write that ", although the 33-47 and 48-62 age groups had equivalent d' scores ($p = .663$)". Instead, please state that "there were no statistically significant differences between d' scores for the 33-47 and 48-62 age groups [full statistics: t , df , p , Cohen's d , CI]" and refrain from further interpretation/discussion of this finding unless this discussion is based on positive evidence for the null.

As a basis for the interpretation of null results, we require Bayes Factors or equivalence tests which should be reported in a separate section in the Results, or next to the current NHST based inference statistics in the Results section.

In your Methods section, please: a) list specified priors and how they were selected; b) describe the statistical model and the techniques used in the analyses; c) summarize the posterior distribution with a measure of central tendency and a credibility interval; d) assess the sensitivity of the analyses to different priors.

Please also include the sensitivity analysis requested by Reviewer #2 (note, we do NOT ask for a post hoc power analysis).

At this stage, we also ask you to publicly deposit the data and code. The code should ideally be DOI-

minted with version control (before resubmission as soon as all analyses are complete). You can find more information below.

Finally, please mention whether you have obtained ethical approval in the manuscript.

Please use the following link to submit your revised manuscript, point-by-point response to the referees' comments (which should be in a separate document to any cover letter) and the completed checklist:

[link redacted]

Please do not hesitate to contact me if you have any questions or would like to discuss these revisions further. We look forward to seeing the revised manuscript and thank you for the opportunity to review your work.

Best regards,

Antonia Eisenkoeck

Antonia Eisenkoeck
Senior Editor
Communications Psychology

EDITORIAL POLICIES AND FORMATTING

Editorial Policy: Policy requirements (Download the link to your computer as a PDF.)

Furthermore, please align your manuscript with our format requirements, which are summarized on the following checklist:

<https://www.nature.com/documents/commspsychol-style-formatting-checklist-article-rr.pdf>>Communications Psychology formatting checklist

and also in our style and formatting guide Communications Psychology formatting guide .

* **CODE AVAILABILITY:** All Communications Psychology manuscripts must include a section titled "Code Availability" at the end of the methods section. In the event of publication, we require that the custom analysis code supporting your conclusions is made available in a publicly accessible repository; at publication, we ask you to choose a repository that provides a DOI for the code; the link to the repository and the DOI will need to be included in the Code Availability statement. Publication as Supplementary Information will not suffice. We ask you to prepare code at this stage, to avoid delays later on in the process.

* **DATA AVAILABILITY:**

All Communications Psychology manuscripts must include a section titled "Data Availability" at the end of the Methods section or main text (if no Methods). More information on this policy, is available at <http://www.nature.com/authors/policies/data/data-availability-statements-data-citations.pdf>><http://www.nature.com/authors/policies/data/data-availability-statements-data-citations.pdf>.

At a minimum the Data availability statement must explain how the data can be obtained and whether there are any restrictions on data sharing. Communications Psychology strongly endorses open sharing of data. If you do make your data openly available, please include in the statement:

We recommend submitting the data to discipline-specific, community-recognized repositories, where possible and a list of recommended repositories is provided at <http://www.nature.com/sdata/policies/repositories>><http://www.nature.com/sdata/policies/repositories>.

If a community resource is unavailable, data can be submitted to generalist repositories such as >figshare or Dryad Digital Repository. Please provide a unique identifier for the data (for example a DOI or a permanent URL) in the data availability statement, if possible. If the repository does not provide identifiers, we encourage authors to supply the search terms that will return the data. For data that have been obtained from publicly available sources, please provide a URL and the specific data product name in the data availability statement. Data with a DOI should be further cited in the methods reference section.

Please refer to our data policies at http://www.nature.com/authors/policies/availability.html.

REVIEWERS' EXPERTISE:

Reviewer #1: misinformation
Reviewer #2: misinformation
Reviewr #3: metacognition

REVIEWERS' COMMENTS:

Reviewer #1 (Remarks to the Author):

This manuscript describes a study that examined the ability to discern true from false headlines, metacognitive certainty in these decisions, and some possible predictors of discernment and metacognitive certainty. Prolific participants decided if each of 140 headlines (70 true and 70 false) were true, rated their certainty in each decision, and completed a few demographic questions. Participants were good at discerning true from false headlines, but Democrats were better at this than Republicans, older participants were better than younger participants, and more educated participants were better than less educated participants. Participants' metacognition was also high, and using only the balanced sample of headlines, Republicans had greater metacognitive efficiency than Democrats. Democrats were also more likely to respond "false" than Republicans across all items. The authors also conducted analyses that included the headlines' political leaning and found some evidence for partisan bias, particularly for Democrats. Finally, there was no evidence for the Dunning-Kruger effect: the lowest performing participants were the most metacognitive efficient.

I enjoyed reading this manuscript. I think that it is important to understand demographic correlates of news discernment and metacognitive certainty. I also appreciate the use of signal detection analyses because the separation of discrimination/discernment from response bias provides a more complete picture of how participants respond to different headlines. I think this manuscript makes a nice contribution to the misinformation literature.

I think this manuscript was well written and clear. My main concerns were addressed in the limitations paragraph: 1) headlines were not balanced for political favorability but they were balanced for truth and 2) the representativeness of older adults on Prolific. I have only a few comments that the authors could address in a revision.

1. A recent article reported some similar findings (<https://doi.org/10.1038/s41598-023-34402-6>). I think the authors should include these findings in their literature review and relate them to their

own findings.

2. As the authors state, participants did well overall in news discernment. Is it possible that the overall performance is a reason that they didn't observe typical Dunning-Kruger effects? That is, with a more difficult headlines, might those performing at the bottom of the distribution show some overconfidence? There is work on the hard-easy effect that suggests that overconfidence occurs on more difficult tasks whereas under-confidence occurs on easier tasks ([https://doi.org/10.1016/0030-5073\(77\)90001-0](https://doi.org/10.1016/0030-5073(77)90001-0)).

3. About 1/3 of the US adult population belongs to neither the Democratic nor Republican parties. I understand that one of the goals of this study was to examine the role of partisanship in news discernment and metacognition, but I think understanding how nonpartisans/independents judge headlines is also important. Another future direction could be to understand independents' discernment ability and metacognitive efficiency. Independents can also provide a control group for partisan biases.

Reviewer #2 (Remarks to the Author):

I enjoyed reading this manuscript investigating how metacognition and partisanship effect fake news discernment. This paper has the potential to make a meaningful contribution to our understanding of the predictors of fake news discernment, namely is demonstrating a null association with metacognition using a more robust measure than past work. Below I lay out some comments in the hopes they help the authors revise the manuscript.

By far my largest suggestion is for greater detail and clarity in the manuscript. I'll preface by acknowledging that the format of Nature-style papers (Intro-Results-Discussion-Method) necessitates that many important details are left until the end. However, currently there is not sufficient information at the end of the introduction for the reader to contextualize the results. Critical information such as the sample size and where the sample was collected need to be stated prior to the results. I also felt like I did not completely understand the design of the study prior to reading the results, for example how Confidence is measured and how metacognitive awareness is computed (and how is the Confidence Likert item transposed into an accurate assessment of one's own accuracy?). Also, while I am relatively familiar with Signal Detection Theory, I worry that a reader who is not will have trouble understanding what d' and m -ratios are, so a better preface of the analysis and SDT is needed.

The results section too is lacking some important details. In general, the results are between-subjects ANOVA, but there are a few analyses that have within-person variables at the stimuli level, do these also control for within-person effects? There are also multiple instances where the authors transform continuous variables into categorical variables (e.g., age, d' , m -ratios). Unless this was preregistered (and I am unable to find a link to the preregistration in the documents I was provided), I view this analytical strategy as statistically suboptimal and requiring explicit justification.

Something that also needs more detail, both as a statistical matter and as a matter of overall inferential validity, is the power/sensitivity of the analyses. In my opinion the primary contribution of this paper is the metacognition findings, which are largely null. As such, a clear picture of the sensitivity of the presented analyses is needed to understand how confident the reader should be in

the inference to the null here. There is brief discussion of statistical power related to the number of trials, but almost all the presented analyses are between-subjects analyses, meaning the number of trials is irrelevant to the power of those analyses (instead, the sample size justification section seems to be referring to the power to confidently detect effect sizes of discernment). Much clearer information is needed about the sensitivity of the presented analyses, and the strength of inferences to the null (possibly using Bayesian methods, if the authors prefer). This then needs to be reflected in the intro/discussion, which overall I think should focus more on the metacognitive findings than the findings which replicate past work.

On that note, I think the introduction/discussion would benefit from a greater focus on the metacognitive findings, in particular how the current design and use of m-ratios (which I believe are a relatively new method, but could be wrong) are superior to past methods. The authors touch on this already, but I think this methodological advance and null findings therein are the primary contribution here, so more space needs to be dedicated to convincing the reader why they should trust these null results over past findings on metacognition and fake news belief. This focus should also replace the parts of the manuscript which merely state that this hasn't been studied before. That's true in general, but better theoretical justification is needed.

One last methodological comment, and this is where my expertise in the presented analyses is strained. There are a few analyses that examined d' and m-ratios in tandem. However, aren't these two variables confounded, since m-ratios are conditioned on d' ? So putting them into a single model together would produce invalid results given the non-independence of the measures? It'd be like taking a difference across of X and Y, then putting X and [X-Y] into a single regression, it violates the independence principle. Like I said, I might be misunderstanding the analyses, or how m-ratios are computed, but this struck me as a potential issue.

I hope the authors find my comments helpful as they revise their manuscript.

Signed Review: Jeff Lees

Reviewer #3 (Remarks to the Author):

I reviewed the paper entitled, "Democrats are better than Republicans at discerning true and false news but do not have better metacognitive awareness."

This paper examines individual differences in the ability to discern true and fake news headlines. They focus on the following individual differences: partisanship, age, education, and gender. They make the following three main predictions: 1. Those who are stronger partisans will exhibit lower levels and metacognitive abilities. 2. Younger adults will be better at discernment and metacognitive abilities, 3. Higher levels of education will predict greater discernment. For exploratory purposes they look at discernment and metacognitive efficiency.

The authors found the following: 1. That Democrats were more discerning than Republicans. They found an interaction effect with partisanship strength. Those who were weaker Republicans were more discerning. However, strong Democrats were more discerning than weak ones. 2. Using politically equated statements did not yield significant differences across age and discernment. 3.

Those who were more educated were better at discernment. The authors then calculate a score of metacognitive efficiency, which can be interpreted to mean how aware a participant is of their ability to discern true from fake news, controlling for performance. They found that Republicans had higher metacognitive efficiency than Democrats and, the stronger partisans were more efficient than the weaker partisans.

In general, I think that this paper takes up an interesting topic and I think that the paper is very well written. However, I feel that the paper's theoretical setup does not sufficiently add to the existing literature. This is a one study paper. I might have liked a study that explained why there might be individual differences in discernment. That is, why might there be differences in metacognitive abilities in different age groups, education levels, etc. Maybe some misinformation is more emotion provoking and certain people are more vulnerable to this. There is already a plethora of research looking at individual differences in discernment across demographics. For partisanship, as you mentioned others have found that Democrats are better at discernment than Republicans (Pennycook & Rand, 2019; Roozenbeek et al., 2022). Likewise, others have already examined differences in discernment and age. There is also research in discernment and intelligence (Ahmed & Tan, 2022; Bago, Rand, & Pennycook, 2020; Pennycook & Rand, 2019; Sanchez & Dunning, 2021). The most novel findings were about metacognitive efficiency. Given that these were done for exploratory purposes, I would have liked to see them preregistered and replicated. To be sure, there is already a literature on politics and expressive responding (2018; Berinsky, 2018; Schaffner, & Luks, 2018).

These are some lesser points:

1. Given that your data contained more false news stories favorable to Republicans, wouldn't you expect to find that Democrats were more accurate than Republicans?
2. Why do you believe that strong Democrats were more discerning compared to weak democrats? I did appreciate it when you looked at the data using politically equated statements.
3. Overall, I found your statistical analysis to be appropriate. However, I would have liked a little more thorough explanation of m-ratio values and how you calculated it. I think I know what you mean but I am not completely certain.
4. Given your data, I am not sure that you can conclude that older adults perform better than younger adults. Why would you predict younger adults will be better at discernment and have better metacognitive abilities? I could also see the opposite prediction or the effect washing out. Others have found that fake news is more likely to be emotional and some have argued that older people have a better memory for emotional information. Maybe older adults experience emotions differently and emotions impact whether people believe in misinformation more than cognitive abilities see: Charles & Carstensen, 2008; Charles, Mather, & Carstensen, 2003; Mather & Carstensen, 2005; Bago, Rosenzweig, Berinsky, & Rand, 2022; Martel, Pennycook, & Rand, 2020; Sanchez & Dunning, 2021; Hess, Popham, Emery, & Elliott, 2012. To be sure, given the correlational nature of this data it is not possible to tell whether it is cognitive decline or motivation.
5. You might have also wanted to recruit some moderates as a control group.

Dear reviewers,

Thank you for the thoughtful review of our manuscript, “Democrats are better than Republicans at discerning true and false news but do not have better metacognitive awareness”, and the opportunity to revise it. We believe the manuscript has greatly improved. Please find below your comments in italics, our responses in plain text, and the changes highlighted in yellow in the manuscript.

Thank you for your continued consideration!

Best regards,

Dr. Briony Swire-Thompson

Reviewer #1 (Remarks to the Author):

This manuscript describes a study that examined the ability to discern true from false headlines, metacognitive certainty in these decisions, and some possible predictors of discernment and metacognitive certainty. Prolific participants decided if each of 140 headlines (70 true and 70 false) were true, rated their certainty in each decision, and completed a few demographic questions. Participants were good at discerning true from false headlines, but Democrats were better at this than Republicans, older participants were better than younger participants, and more educated participants were better than less educated participants. Participants’ metacognition was also high, and using only the balanced sample of headlines, Republicans had greater metacognitive efficiency than Democrats. Democrats were also more likely to respond “false” than Republicans across all items. The authors also conducted analyses that included the headlines’ political leaning and found some evidence for partisan bias, particularly for Democrats. Finally, there was no evidence for the Dunning-Kruger effect: the lowest performing participants were the most metacognitive efficient.

I enjoyed reading this manuscript. I think that it is important to understand demographic correlates of news discernment and metacognitive certainty. I also appreciate the use of signal detection analyses because the separation of discrimination/discernment from response bias provides a more complete picture of how participants respond to different headlines. I think this manuscript makes a nice contribution to the misinformation literature.

Thank you very much!

I think this manuscript was well written and clear. My main concerns were addressed in the limitations paragraph: 1) headlines were not balanced for political favorability but they were balanced for truth and 2) the representativeness of older adults on Prolific. I have only a few comments that the authors could address in a revision.

1. A recent article reported some similar findings (<https://doi.org/10.1038/s41598-023-34402-6>). I think the authors should include these findings in their literature review and relate them to their own findings.

Thank you for highlighting this relevant paper, we have now integrated it in several places throughout our paper. For example, see p. 4, “Finally, Arin et al.¹³ concluded that people have a good assessment of their ability to detect false news, but they only compared whether participants would hypothetically share false news with whether they had shared false news in the past. In other words, their study focused on sharing rather than explicit measures of discernment ability”. We also cite it when referring to differences in discernment in partisanship, age, gender, and education.

2. As the authors state, participants did well overall in news discernment. Is it possible that the overall performance is a reason that they didn’t observe typical Dunning-Kruger effects? That is, with more difficult headlines, might those performing at the bottom of the distribution show some overconfidence? There is work on the hard-easy effect that suggests that overconfidence occurs on more difficult tasks whereas under-confidence occurs on easier tasks ([https://doi.org/10.1016/0030-5073\(77\)90001-0](https://doi.org/10.1016/0030-5073(77)90001-0)).

Thank you for pointing this out. We have now added this to our limitations section, p. 24 “Another possible reason that we did not observe typical Dunning-Kruger effects is because our discernment task was relatively easy and this miscalibration has been shown to increase with the difficulty of the task (Lichtenstein & Fischhoff, 1977)”. We have also added the citation you suggested.

3. About 1/3 of the US adult population belongs to neither the Democratic nor Republican parties. I understand that one of the goals of this study was to examine the role of partisanship in news discernment and metacognition, but I think understanding how nonpartisans/independents judge headlines is also important. Another future direction could be to understand independents’ discernment ability and metacognitive efficiency. Independents can also provide a control group for partisan biases.

We absolutely agree, and have added this as a future direction on p. 25 “Finally, we did not recruit political independents or non-partisans, despite the fact that a large proportion of U.S. adults identify as such (Gallup, 2023), and future research could include people who don’t identify with the Democratic or Republican parties”.

Reviewer #2 (Remarks to the Author):

I enjoyed reading this manuscript investigating how metacognition and partisanship affect fake news discernment. This paper has the potential to make a meaningful contribution to our understanding of the predictors of fake news discernment, namely in demonstrating a null association with metacognition using a more robust measure than past work. Below I lay out some comments in the hopes they help the authors revise the manuscript.

Thank you!

By far my largest suggestion is for greater detail and clarity in the manuscript. I'll preface by acknowledging that the format of Nature-style papers (Intro-Results-Discussion-Method) necessitates that many important details are left until the end. However, currently there is not sufficient information at the end of the introduction for the reader to contextualize the results. Critical information such as the sample size and where the sample was collected need to be stated prior to the results. I also felt like I did not completely understand the design of the study prior to reading the results, for example how Confidence is measured and how metacognitive awareness is computed (and how is the Confidence Likert item transposed into an accurate assessment of one's own accuracy?). Also, while I am relatively familiar with Signal Detection Theory, I worry that a reader who is not will have trouble understanding what d' and m -ratios are, so a better preface of the analysis and SDT is needed.

We agree, and now believe that the new order recommended by the editor (*Introduction - Methods - Results - Discussion*), is much clearer for readers.

The results section too is lacking some important details. In general, the results are between-subjects ANOVA, but there are a few analyses that have within-person variables at the stimuli level, do these also control for within-person effects? There are also multiple instances where the authors transform continuous variables into categorical variables (e.g., age, d' , m -ratios). Unless this was preregistered (and I am unable to find a link to the preregistration in the documents I was provided), I view this analytical strategy as statistically suboptimal and requiring explicit justification.

Given that multiple reviewers missed that our study was pre-registered, we have now added a "pre-registration" section prior to the results to emphasize this. Please find our pre-registration link here: <https://osf.io/ay9fc/>.

We additionally clarify the details suggested in the results section. First, our ANOVAs that have within-subjects factors are between-within (repeated-measures) which control for within-person variables. Second, we justify transforming continuous variables to categorical variables. We write "We chose these age bins to replicate previous work showing that people aged 18-30 differed in comparison to people aged 65+ in fake news engagement online (Grinberg et al., 2019; Guess et al., 2019)." Our 18-32, 33-47, 48-62, 63+ age splits are indeed pre-registered groupings. All analyses are conducted using the continuous variables as well. For example, "For robustness, we also correlated d' with age, and found that they were positively associated, albeit modestly ($\rho = .11$, $p = .016$)."

Something that also needs more detail, both as a statistical matter and as a matter of overall inferential validity, is the power/sensitivity of the analyses. In my opinion the primary contribution of this paper is the metacognition findings, which are largely null. As such, a clear picture of the

sensitivity of the presented analyses is needed to understand how confident the reader should be in the inference to the null here. There is brief discussion of statistical power related to the number of trials, but almost all the presented analyses are between-subjects analyses, meaning the number of trials is irrelevant to the power of those analyses (instead, the sample size justification section seems to be referring to the power to confidently detect effect sizes of discernment). Much clearer information is needed about the sensitivity of the presented analyses, and the strength of inferences to the null (possibly using Bayesian methods, if the authors prefer). This then needs to be reflected in the intro/discussion, which overall I think should focus more on the metacognitive findings than the findings which replicate past work.

We now include sensitivity analyses on the metacognition measures in the sample size justification section (see pages 8-9): “A sensitivity analysis in G*Power indicated that our final sample size of 500 people had 95% power to detect the outcomes of a between-subjects ANOVAs for our partisanship and partisanship × partisanship strength analyses on discernment and metacognitive ability of at least $f = .19$, and one-way ordinal ANOVAs with age group as a factor at $f = .19$. This aligns with recommendations by Brysbaert (2019), recommending that $f = .20$ is a good estimate for the smallest effect size of interest in psychological research. We nonetheless repeat all analyses with Bayesian methods, reporting all null findings with the relative evidence favoring the null (BF_{01}) quantified in the main text. For reference, a BF between 1-3 providing anecdotal evidence, 3-10 moderate evidence, 10-30 strong evidence, 30-100 very strong evidence, and a BF greater than 100 constitutes extreme evidence (Wagenmakers et al., 2018).”

On that note, I think the introduction/discussion would benefit from a greater focus on the metacognitive findings, in particular how the current design and use of m-ratios (which I believe are a relatively new method, but could be wrong) are superior to past methods. The authors touch on this already, but I think this methodological advance and null findings therein are the primary contribution here, so more space needs to be dedicated to convincing the reader why they should trust these null results over past findings on metacognition and fake news belief. This focus should also replace the parts of the manuscript which merely state that this hasn't been studied before. That's true in general, but better theoretical justification is needed.

We now add several sentences emphasizing the novelty of our measurement approach to the field of misinformation, for example, “A notable strength of this study is its use of the *meta-d'* framework to measure metacognitive ability (Maniscalco & Lau, 2012; Fleming, 2017). This framework has several advantages over and above approaches used in other misinformation studies, including removing performance and response bias artifacts and quantifying discernment ability and response bias separately (Fleming & Lau, 2014)”.

One last methodological comment, and this is where my expertise in the presented analyses is strained. There are a few analyses that examined d' and m-ratios in tandem. However, aren't these two variables confounded, since m-ratios are conditioned on d' ? So putting them into a

single model together would produce invalid results given the non-independence of the measures? It'd be like taking a difference across of X and Y, then putting X and [X-Y] into a single regression, it violates the independence principle. Like I said, I might be misunderstanding the analyses, or how m-ratios are computed, but this struck me as a potential issue.

This is an excellent point. To ensure the variables are not too overlapping, we correlated them and found that d' and m-ratio were not positively correlated ($r = -.20$). We only put them in the same model during one analysis, when examining whether the least accurate participants maintained the least insight into their ability. Here we split d' and m-ratio into even and odd trials to compare the two.

I hope the authors find my comments helpful as they revise their manuscript.

We appreciate the helpful comments!

Signed Review: Jeff Lees

Reviewer #3 (Remarks to the Author):

I reviewed the paper entitled, "Democrats are better than Republicans at discerning true and false news but do not have better metacognitive awareness."

This paper examines individual differences in the ability to discern true and fake news headlines. They focus on the following individual differences: partisanship, age, education, and gender. They make the following three main predictions: 1. Those who are stronger partisans will exhibit lower levels and metacognitive abilities. 2. Younger adults will be better at discernment and metacognitive abilities, 3. Higher levels of education will predict greater discernment. For exploratory purposes they look at discernment and metacognitive efficiency.

The authors found the following: 1. That Democrats were more discerning than Republicans. They found an interaction effect with partisanship strength. Those who were weaker Republicans were more discerning. However, strong Democrats were more discerning than weak ones. 2. Using politically equated statements did not yield significant differences across age and discernment. 3. Those who were more educated were better at discernment. The authors then calculate a score of metacognitive efficiency, which can be interpreted to mean how aware a participant is of their ability to discern true from fake news, controlling for performance. They found that Republicans had higher metacognitive efficiency than Democrats and, the stronger partisans were more efficient than the weaker partisans.

In general, I think that this paper takes up an interesting topic and I think that the paper is very well written.

Thank you!

However, I feel that the paper's theoretical setup does not sufficiently add to the existing literature. This is a one study paper. I might have liked a study that explained why there might be individual differences in discernment. That is, why might there be differences in metacognitive abilities in different age groups, education levels, etc. Maybe some misinformation is more emotion provoking and certain people are more vulnerable to this. There is already a plethora of research looking at individual differences in discernment across demographics. For partisanship, as you mentioned others have found that Democrats are better at discernment than Republicans (Pennycook & Rand, 2019; Roozenbeek et al., 2022). Likewise, others have already examined differences in discernment and age. There is also research in discernment and intelligence (Ahmed & Tan, 2022; Bago, Rand, & Pennycook, 2020; Pennycook & Rand, 2019; Sanchez & Dunning, 2021). The most novel findings were about metacognitive efficiency. Given that these were done for exploratory purposes, I would have liked to see them preregistered and replicated. To be sure, there is already a literature on politics and expressive responding (Berinsky, 2018; Schaffner, & Luks, 2018).

Thank you for this feedback. Given that multiple reviewers missed that our study was pre-registered, we have now added a “pre-registration” section prior to the results to emphasize this. Please find our pre-registration link here: <https://osf.io/ay9fc/>. We note that we did preregister hypotheses regarding metacognitive efficiency, and thus they were not exploratory.

While we agree that there is significant previous research on demographics differences in discernment, the main contribution of the present manuscript is metacognition, which we now better emphasize in the introduction (see page 6). In terms of discernment, we consider reasons for these demographic differences (such as emotion) an exciting direction for further research and include this in the discussion. We now write “Future studies should consider why such age differences exist (e.g., differences in emotional processing; Charles & Carstensen, 2008; Bago et al., 2022), particularly considering that older adults share more misinformation online than younger adults (see Brashier & Schacter, 2020)”.

These are some lesser points:

1. Given that your data contained more false news stories favorable to Republicans, wouldn't you expect to find that Democrats were more accurate than Republicans?

This is a reasonable hypothesis. However, we found that even when we equated the items for political favorability, Democrats were still more accurate than Republicans. We include the fact that our larger stimuli set was not balanced for political favorability between true and false items as a limitation.

2. Why do you believe that strong Democrats were more discerning compared to weak democrats? I did appreciate it when you looked at the data using politically equated statements.

Yes this is indeed an interesting finding, particularly because while strong Democrats were *more* discerning, strong Republicans were *less* discerning, replicating a finding by Garrett and Bond (2021). We now add in the discussion on p. 22-23, “One potential reason for this is that strong partisans on both sides are more likely to engage with a greater quantity of news, but Democrats are more likely to engage with a greater *breadth* of news sources than Republicans (Mitchell et al., 2021), contributing to different prior beliefs about what is or is not plausible (Pennycook & Rand, 2019)”.

3. Overall, I found your statistical analysis to be appropriate. However, I would have liked a little more thorough explanation of m-ratio values and how you calculated it. I think I know what you mean but I am not completely certain.

We are happy to add further information. The m-ratio is calculated by dividing meta d' by d'. On page 12, we write, “Simply dividing meta d' by d' removes type-1 performance artifacts, producing a measure of metacognitive efficiency called the m-ratio. M-ratio values can be interpreted as a participant's metacognitive ability given a specific level of task performance, or how metacognitively capable (efficient) one is given how difficult one finds the task. A metacognitively ideal observer would produce a meta-d' value equivalent to their d' value, yielding an m-ratio of 1.” We have additional information on calculating meta-d' and m-ratio values in the Supplementary Methods. Please let us know if there is anything specific that you would like to see in the main text.

4. Given your data, I am not sure that you can conclude that older adults perform better than younger adults. Why would you predict younger adults will be better at discernment and have better metacognitive abilities? I could also see the opposite prediction or the effect washing out. Others have found that fake news is more likely to be emotional and some have argued that older people have a better memory for emotional information. Maybe older adults experience emotions differently and emotions impact whether people believe in misinformation more than cognitive abilities see: Charles & Carstensen, 2008; Charles, Mather, & Carstensen, 2003; Mather & Carstensen, 2005; Bago, Rosenzweig, Berinsky, & Rand, 2022; Martel, Pennycook, & Rand, 2020; Sanchez & Dunning, 2021; Hess, Popham, Emery, & Elliott, 2012. To be sure, given the correlational nature of this data it is not possible to tell whether it is cognitive decline or motivation.

We back off the strong claim that older adults outperformed younger adults and note that “However, it is important to note that our effect sizes are quite small, and might be specific to the current paradigm, and should be replicated”. We also add that future studies should consider why such age differences potentially exist (e.g., differences in emotional processing; Charles & Carstensen, 2008; Bago et al., 2022), particularly considering that older adults share more misinformation online than younger adults (see Brashier & Schacter, 2020)”.

We pre-registered that younger adults would be better at metacognitive abilities because metacognitive abilities generally decrease as people get older (Palmer, David, &

Fleming, 2014; Culot et al., 2022). However, we also agree that our prediction regarding younger adults being better at discernment is not well supported by the current literature, and now state “Note that we pre-registered that older adults would score lower than younger adults. However, we subsequently realized that this did not align well with the current literature, and predictions of older adults being better at discernment would have been better aligned.”

5. You might have also wanted to recruit some moderates as a control group.

Thank you. We now include this as a limitation on p. 25, “Finally, we did not recruit political independents or non-partisans, despite the fact that a large proportion of U.S. adults identify as such (Gallup, 2023), and future research should include people who don’t identify with the Democratic or Republican parties”.

26th Oct 23

Dear Dr Swire-Thompson,

Your manuscript titled "Democrats are better than Republicans at discerning true and false news but do not have better metacognitive awareness" has now been seen by our reviewers, whose comments appear below. In light of their advice I am delighted to say that we are happy, in principle, to publish a suitably revised version in Communications Psychology under the open access CC BY license (Creative Commons Attribution v4.0 International License).

We therefore invite you to revise your paper one last time to address the remaining concerns of our reviewers and a list of editorial requests. At the same time we ask that you edit your manuscript to comply with our format requirements and to maximise the accessibility and therefore the impact of your work.

Please note that it may still be possible for your paper to be published before the end of 2023, but in order to do this we will need you to address these points as quickly as possible so that we can move forward with your paper.

EDITORIAL REQUESTS:

Please also discuss in your limitations section that $f=0.2$ may not be a suitably small effect size to rule out differences that some might consider of theoretical or practical importance (see Reviewer #2's comments).

SUBMISSION INFORMATION:

OPEN ACCESS:

Communications Psychology is a fully open access journal. Articles are made freely accessible on publication under a [CC BY](http://creativecommons.org/licenses/by/4.0) license (Creative Commons Attribution 4.0 International License). This license allows maximum dissemination and re-use of open access materials and is preferred by many research funding bodies.

For further information about article processing charges, open access funding, and advice and

support from Nature Research, please visit <https://www.nature.com/commpsychol/article-processing-charges>

At acceptance, you will be provided with instructions for completing this CC BY license on behalf of all authors. This grants us the necessary permissions to publish your paper. Additionally, you will be asked to declare that all required third party permissions have been obtained, and to provide billing information in order to pay the article-processing charge (APC).

* **DATA AVAILABILITY:**

[link redacted]

Best regards,

Antonia Eisenkoeck

Antonia Eisenkoeck
Senior Editor
Communications Psychology

REVIEWERS' COMMENTS:

Reviewer #1 (Remarks to the Author):

The authors have carefully and completely addressed my previous concerns. I have no new concerns and believe this manuscript makes a nice contribution to the literature.

Reviewer #2 (Remarks to the Author):

Overall, I find the manuscript much improved, and ready for publication! I have one tiny reporting request, and lastly an open thought the authors can do with what they wish.

Reporting wise, I would ask that the authors include clear information regarding how “confidence” is measured and used. I think it’s a Likert scale that they medians-split to determine accuracy (a la Table 1), but I’m not sure. The scale that Confidence is on, and precisely how it’s used to determine accuracy, should be clearer.

Regarding the open thought...in the updated power analyses the authors cite a Brysbaert (2019) claiming that an effect size of $f=0.2$ (i.e, $d = 0.4$) is the smallest effect size of interest in psychology. On a personal level (like I said, the authors should feel free to do with my opinion what they wish), I find Brysbaert’s claim baffling. Usually, I hear $f=0.1$ cited as the smallest of interest, and even then I think it should be closer to $f=0.05$. Just to list of few real-world effect sizes smaller than $f=0.2$: the association between high school grades and job performance; the effect of lead exposure on children’s IQ levels; the effect of maternal alcohol use on premature birth risk; the effect of parental divorce on children’s wellbeing; the gender gap in risk-taking behaviors; the effect of ibuprofen on pain. Like I said, just my opinion...no need to remove the citation or change the sensitivity analyses. But insofar as my opinion is useful norm information, when I’m preparing a study for which I lack strong priors regarding the anticipated effect size, I power my studies to detect an effect of $f=0.1$.

Congratulations on a job well done!

Signed review: Jeff Lees

Reviewer #3 (Remarks to the Author):

I rereviewed the paper entitled, “Democrats are better than Republicans at discerning true and false news but do not have better metacognitive awareness.” Overall, I found the manuscript much improved. Also, I believe that the authors have addressed the major concerns that the editor noted

from the initial manuscript. Specifically, they added more details regarding their statistical models. Further clarifications and justification for their predictions and design, and cited additional literature. They have also added that their paper was preregistered and provided justification for their sample. The authors have addressed theoretical concerns identified by the editor.

Dear reviewers,

Thank you for the thoughtful review of our manuscript, “Democrats are better than Republicans at discerning true and false news but do not have better metacognitive awareness”, and the opportunity to revise it. Please find below your comments in italics, our responses in plain text, and the changes highlighted in yellow in the manuscript.

Thank you for your continued consideration!

Best regards,

Dr. Briony Swire-Thompson

Reviewer #1 (Remarks to the Author):

The authors have carefully and completely addressed my previous concerns. I have no new concerns and believe this manuscript makes a nice contribution to the literature.

Thank you very much!

Reviewer #2 (Remarks to the Author):

Overall, I find the manuscript much improved, and ready for publication! I have one tiny reporting request, and lastly an open thought the authors can do with what they wish.

Reporting wise, I would ask that the authors include clear information regarding how “confidence” is measured and used. I think it’s a Likert scale that they medians-split to determine accuracy (a la Table 1), but I’m not sure. The scale that Confidence is on, and precisely how it’s used to determine accuracy, should be clearer.

We have added additional details about how confidence was measured and used to determine accuracy. We now write on p.12: “In this study, confidence was rated on a four-point scale. Ratings of three or four were considered high confidence, and confidence ratings of one or two were considered low confidence.”

Regarding the open thought...in the updated power analyses the authors cite a Brysbaert (2019) claiming that an effect size of $f=0.2$ (i.e, $d = 0.4$) is the smallest effect size of interest in psychology. On a personal level (like I said, the authors should feel free to do with my opinion what they wish), I find Brysbaert’s claim baffling. Usually, I hear $f=0.1$ cited as the smallest of interest, and even then I think it should be closer to $f=0.05$. Just to list of few real-world effect sizes smaller than $f=0.2$: the association between high school grades and job performance; the effect of lead exposure on children’s IQ levels; the effect of maternal alcohol use on premature birth risk; the effect of parental divorce on children’s wellbeing; the gender gap in risk-taking behaviors; the effect of ibuprofen on pain. Like I said, just my opinion...no need to remove the citation or change the sensitivity analyses. But insofar as my opinion is useful norm information,

when I'm preparing a study for which I lack strong priors regarding the anticipated effect size, I power my studies to detect an effect of $f=0.1$.

We agree and have added language about how effect sizes smaller than $f = 0.2$ can still be meaningful on p. 9: "However, we acknowledge that we may be underpowered to detect smaller effect sizes. We therefore repeat all analyses with Bayesian methods.."

Congratulations on a job well done!

Signed review: Jeff Lees

Reviewer #3 (Remarks to the Author):

I rereviewed the paper entitled, "Democrats are better than Republicans at discerning true and false news but do not have better metacognitive awareness." Overall, I found the manuscript much improved. Also, I believe that the authors have addressed the major concerns that the editor noted from the initial manuscript. Specifically, they added more details regarding their statistical models. Further clarifications and justification for their predictions and design, and cited additional literature. They have also added that their paper was preregistered and provided justification for their sample. The authors have addressed theoretical concerns identified by the editor.

Thank you!